# Plastome Evolution and Comparative Analyses of a Recently Radiated Genus *Vanda* (Aeridinae, Orchidaceae)

**DOI:** 10.3390/ijms25179538

**Published:** 2024-09-02

**Authors:** Wanshun Lei, Peng Zhou, Zelong Pei, Yizhen Liu, Yan Luo, Xiaoguo Xiang

**Affiliations:** 1Key Laboratory of Poyang Lake Environment and Resource Utilization, Ministry of Education, School of Life Sciences, Nanchang University, Nanchang 330031, China; 405600220103@email.ncu.edu.cn (W.L.); 355600230021@email.ncu.edu.cn (P.Z.); 5604120024@email.ncu.edu.cn (Z.P.); liuyizhen@ncu.edu.cn (Y.L.); 2Xishuangbanna Tropical Botanical Garden, Chinese Academy of Sciences, Mengla 666300, China; luoyan@xtbg.org.cn

**Keywords:** *Vanda*, plastome, mutational hotspots, phylogenomics, rapid radiation lineages

## Abstract

*Vanda* R.Br. is an epiphytic orchid genus with significant horticultural and ornamental value. Previous molecular studies expanded *Vanda* including some members from five other genera. However, the interspecific relationships of this recently radiated genus have remained unclear based on several DNA markers until now. In this study, the complete plastome has been used to infer the phylogenetic relationships of *Vanda s.l*. The five newly obtained plastomes ranged from 146,340 bp to 149,273 bp in length, with a GC content ranging from 36.5% to 36.7%. The five plastomes contained 74 protein-coding genes (CDSs), 38 tRNAs, and 8 rRNAs, and their *ndh* genes underwent loss or pseudogenization. Comparative plastome analyses of 13 *Vanda* species revealed high conservation in terms of genome size, structure, and gene order, except for a large inversion from *trnG^GCC^* to *ycf3* in *V. coerulea*. Moreover, six CDSs and five non-CDSs were selected as candidate DNA barcodes. Our phylogenetic analyses demonstrated that *Vanda s.l.* is a monophyletic group with high supporting values based on five different datasets (complete plastome with one IR, 68 CDSs, LSC, five hypervariable non-CDSs, and six hypervariable CDSs), while the phylogenetic relationships among species were fully resolved based on the complete plastome with one IR dataset. Our results confirmed that the complete plastome has a great power in resolving the phylogenetic relationships of recently radiated lineages.

## 1. Introduction

*Vanda* R.Br. (Aeridinae, Vandeae, Epidendroideae, and Orchidaceae) comprises approximately 73 species [1,2], and is found in subtropical and tropical Asia and Australia (Queensland). Over half of the species exhibit narrow endemism, with a center of species diversity in the South-East Asian archipelagos [3]. Particularly, there is great diversity in floral characters, and thus, *Vanda* has been recognized as one of the five most horticulturally important orchid genera in the world [3,4,5].

*Vanda* is considered to be one of the most taxonomically complicated groups in Aeridinae, due to its morphological similarity to *Ascocentrum*, *Euanthe*, *Trudelia*, and *Christensonia* [4]. The genus *Vanda* was established in 1820 by Robert Brown [6]. Lindley [7] redefined this genus, and postulated five sections. Gardiner [8] transferred 17 species from *Ascocentrum*, *Ascocentropsis*, *Christensonia*, *Eparmatostigma*, and *Neofinetia* into *Vanda*, and proposed to establish *Vanda s.l*. Subsequently, Gardiner et al. [3] used three plastid regions (*matK*, *psbA*-*trnH*, and *trnL*-*F*) to partly resolve the phylogenetic relationships among *Vanda s.l*., and proposed 13 morphological sectional classifications. Chase et al. [2] adopted this broad taxonomic generic concept. Although the *Vanda s.l*. has been widely accepted, the infrageneric relationships remained controversial in phylogenetic research. For example, *V. alpina* and *V. pumila* were clustered together based on five DNA sequences (ITS, *atpI-H*, *matK*, *psbA-trnH*, and *trnL-F*) [9], but this relationship was not supported when using ITS and five plastid DNA regions (*atpI*-*H*, *matK*, *psbA*-*trnH*, *trnL*-*F*, and *trnS*-*trnfM*) [10]. Moreover, many sections proposed by Gardiner et al. [3] were not supported as monophyletic groups based on ITS and three plastid DNA regions (*matK*, *trnH*-*psbA*, and *trnL*-*F*) [11]. Zhang et al. [12] investigated the diversification of Orchidaceae and inferred that *Vanda s.l.* originated 8.19 million years ago (Ma), and then rapidly diversified following the Pliocene. Briefly, several DNA markers had limited power to reveal the interspecific relationships of the radiated *Vanda s.l*. Therefore, it is crucial to explore more genetically varied loci to elucidate the phylogenetic relationships within *Vanda*.

Until now, the complete plastome has successfully clarified the phylogenetic relationships of recently radiated lineages. For instance, Li et al. [13] used plastome data to explore the relationships within *Holcoglossum*, which had divergently radiated since the late Miocene [14]. Moreover, the phylogenetic relationships of *Eriocaulon*, which diverged following the late Miocene and diversified in the Quaternary, were well resolved by the complete plastome [15]. Wang et al. [16] successfully verified the species relationships within *Lagerstroemia* (Lythraceae) based on the plastome data, which rapidly radiated following the late Miocene. Thus, it is necessary to explore the plastome character of *Vanda s.l.* and employ the complete plastome to investigate these recently rapidly diverging lineages.

In this study, the plastomes of five *Vanda* species were newly obtained, and eight published plastomes were downloaded from GenBank. In total, 13 plastomes were combined to conduct plastome comparative analyses and phylogenetic reconstruction within *Vanda s.l*. The goals of this study are: (1) to reveal the gene content, gene order, and structures of *Vanda* plastomes; (2) to detect the hypervariable regions of these plastomes as prospective DNA markers for species identification; (3) to explore the phylogenetic relationships within this genus. Our study provides valuable genetic information on the phylogeny, species identification, and conservation of *Vanda s.l*.

## 2. Result

### 2.1. The Plastome Characters of Vanda

The plastomes of *Vanda* species contained four regions: a large single copy (LSC) region, a small single copy (SSC) region, and two copies of inverted repeats (IRa/b) (Figure 1). The total length of the 13 plastomes in the genus *Vanda* ranged from 146,340 bp (*V. ampullaceum*) to 149,490 bp (*V. subconcolor*) (Table 1). The lengths of the LSC regions were 83,808–85,982 bp, those of the SSC regions were 11,424–12,002 bp, and a pair of IR regions ranged from 24,523 to 25,965 bp (Table 1). The total GC contents ranged from 36.5% to 36.7%. All of the 13 *Vanda* plastomes contained 120 genes, comprising 74 protein-coding genes (CDSs), 38 transfer RNA (tRNA) genes, and eight ribosomal RNA (rRNA) genes (Table 1).

All the *ndh* genes of *Vanda* plastomes had been lost or pseudogenized (Figure 1; Table 1). The plastomes of *V. cristata*, *V. falcata*, *V. richardsiana*, and *V. xichangensis* possessed five pseudogenes (*ndhB*, *ndhD*, *ndhE*, *ndhG*, and *ndhI*) and lost six genes (*ndhA*, *ndhC*, *ndhF*, *ndhH*, *ndhJ*, *and ndhK*). The other nine species possessed eight pseudogenes (*ndhB*, *ndhC*, *ndhD*, *ndhE*, *ndhG*, *ndhI*, *ndhJ*, and *ndhK*), and lost three genes (*ndhA*, *ndhF*, and *ndhH*) (Figure 1).

There was no expansion and contraction in the IR/SC boundary regions across the 13 species of *Vanda* (Appendix A). *Rpl22* (318–329 bp) genes of the LSC crossed over into IRb. The *trnN* (72 bp) and *rpl32* (174 bp) genes were near to the junction between SSC and IRb. Within the junction region between SSC and IRa, the *ycf1* genes were across the SSC-IRa boundary, and primarily located in the SSC region, with lengths ranging from 5211 bp (*V. coerulescens*) to 5328 bp (*V. concolor*). The adjacent regions of LSC and IRb were located in the *rps19* (279 bp) genes and *psbA* (1062 bp) genes. The collinearity analysis revealed that the SSC and IR regions of 13 *Vanda* plastomes were highly conserved. Additionally, conservation was observed in the LSC region of 12 *Vanda* plastomes, and a significant inversion from *trnG^GCC^* to *ycf3* was identified in the LSC region of the *V*. *coerulea* plastome (Figure 2).

### 2.2. Repeated Analysis

Different types of the simple sequence repeats (SSRs) of *Vanda* were detected (Figure 3A,B). The number of identified SSRs ranged from 52 (*V. coerulea*) to 69 (*V. subconcolor*). There were 775 SSRs detected in the 13 *Vanda* plastomes. Six types of SSRs (mono-nucleotide, di-nucleotide, tri-nucleotide, tetra-nucleotide, penta-nucleotide, and hexa-nucleotide) were identified, with 514 SSRs (66.32%) being mono-nucleotide type, particularly A and T repeat motifs. In di-nucleotide repeat type (115), the number of AT/TA repeat motifs (89) exceeded that of TC/GA (26). Hexa-nucleotide repeats (3) occurred with the lowest frequency, which was only detected in *V. coerulescens*, *V. concolor*, and *V. subconcolor*. Additionally, the LSC region had a higher content of SSRs (530) than the SSC region (131) and the IR regions (114). There were 637 long repeats in the *Vanda* plastomes, of which 213 (33.44%) were forward repeats, 324 (50.86%) were palindromic repeats, 96 (15.07%) were reverse repeats, and four (0.63%) were complement repeats. The lengths of the long repeats mainly ranged from 20 bp to 39 bp.

### 2.3. Codon Usage Bias

The codon usage frequency of *Vanda* plastomes was calculated based on concatenated sequences of 68 CDSs. The heatmap showed that codon usage bias was highly conserved in *Vanda* (Appendix A). The relative synonymous codon usage (RSCU) analysis demonstrated that codons containing A/T at 3′ exhibited high RSCU values (RSCU ≥ 1), and codons ending with C/G at 3′ showed low RSCU values (RSCU < 1). The RSCU values of *Vanda* plastomes ranged from 0.29 to 1.89 (Appendix A). The AGA codon had the highest RSCU values (1.86–1.89), while AGC exhibited the lowest RSCU values (0.29–0.30). The UGA codon had higher RSCU values (0.66–0.70) than the other two termination codons (UAA and UAG). In addition, leucine (Leu) had the highest count of codons (2062–2086), while cysteine (Cys) exhibited the lowest (226–229).

### 2.4. Plastome Sequence Divergence and DNA Marker Investigation

The structural characteristics of 13 *Vanda* plastomes were investigated to evaluate the sequence of the plastomes (Appendix A). The coding regions exhibited higher conservation than the non-coding regions, and the LSC and SSC regions displayed greater variability than the IR regions (Appendix A). The nucleotide diversity (Pi) was calculated to identify the highly mutated hotspots of *Vanda* plastomes (Figure 4). The results showed that the Pi values of the LSC region, SSC region, and IR regions were 0.0075, 0.0064, and 0.0016, respectively. Based on the ranking of the Pi values, five hypervariable regions, including *clpP*-*psbB*, *rps15-ycf1*, *trnN^GUU^*-*rpl32*, *trnS^GCU^*-*trnG^GCC^*, and *trnT^UGU^*-*trnL^UAA^* were identified in the complete plastomes. As for 68 CDSs, six hypervariable regions were identified, including *ccsA*, *psbI*, *rpl22*, *rpoA*, *rps15*, and *ycf1*.

### 2.5. Phylogenetic Analyses and Character Reconstruction

Maximum likelihood (ML) and Bayesian inference (BI) analyses yielded nearly identical topologies based on the complete plastome with one IR, 68 CDSs, LSC, five hypervariable non-CDSs and six hypervariable CDSs, respectively, except the uncertain systematic position of *V. ampullaceum* (Figure 5 and Appendix A). The monophyly of *Vanda s.l.* have confirmed by the five datasets with high Bootstrap percentages (BS values) from the ML analyses and Posterior Probability (PP values) from the Bayesian analyses, respectively. The phylogenetic relationships among species were fully resolved based on the complete plastome (excluding one IR) matrix, and the phylogenetic tree was illustrated here. *Vanda s.l.* was well supported as a monophyletic group (PP-BI = 1.00, BS-ML = 100%) (Figure 5). Three species from sect. *Neofinetia* formed a clade and diverged first (PP-BI = 1.00, BS-ML = 100%). The other sections formed another clade with high to moderate supporting values (PP-BI = 1.00, BS-ML = 71%). The three species from sect. *Cristatae* and the two species from sect. *Longicalcarata* did not form a clade.

The reconstruction of *ndh* gene status showed that the most recent common ancestor (MRCA) of *Vanda* compared to its relatives involved the *ndhI* pseudogene. The *ndhC*, *ndhJ*, and *ndhK* were lost in sect. *Neofinetia* and *V. cristata* (Figure 5).

## 3. Discussion

### 3.1. The Plastome Characteristics and Structural Evolution

In this study, the complete plastome sequences of five *Vanda* species were newly obtained. The *Vanda* plastomes contained one LSC, one SSC, and two IR regions (Figure 1), which is similar to most angiosperm [13,15,16]. The genome sizes ranged from 146,340 bp to 149,490 bp and the GC contents of the plastomes range from 36.5% to 36.7% and fall within the range of other orchids, such as *Aerides* and *Angraecum* [17,18]. A total of 120 genes were identified in the *Vanda* plastomes, consisting of 74 CDSs, 38 tRNA genes, and 8 rRNA genes (Table 1). Further, all *ndh* genes were pseudogenized or lost (Figure 5). The *ndh* genes have been involved in photosynthesis, the photosynthetic response and stress acclimation, and have been hypothesized to be closely related to plant transition to terrestrial habitats and are usually absent in epiphytic plants [19,20]. Previous studies showed that all 11 *ndh* genes were lost or pseudogenized in nearly all photosynthetic and non-photosynthetic orchids, which suggests that the *ndh* genes have been independently lost in various orchid lineages [21]. As such, all of the *ndh* genes were truncated or pseudogenized in the epiphytic orchid genera *Holcoglossum* [13], *Paraphalaenopsis* [22], and *Trichoglottis* [23].

Our results reveal that the gene arrangement of the IR/SC boundary in *Vanda* plastomes was extremely conserved (Appendix A), and the gene content of this genus was identical. This finding implied that the length variation of *Vanda* plastomes did not attribute to the IR/SC boundary shift, although the IR/SC boundary shift has been considered to be the main factor for the variations in plastome length and gene content [24].

Previous studies have revealed that gene structure and gene order are highly conserved in orchid plastomes [25,26]. Our collinearity analysis revealed that the structures of 13 *Vanda* plastomes were conserved, while there was a large inversion from *trnG^GCC^* to *ycf3* existing in the *V. coerulea* plastome (Figure 2). Plastome rearrangement was also reported for the plastome of Orchidaceae. For example, *Cypripedium formosanum* has an inversion from *atpA* to *petG* [27], and there is a large inversion from *trnS^GCU^* to *trnS^GGA^* in the *Apostasia wallichii* plastome [28].

Our results showed that among 13 *Vanda* plastomes, the codon usage bias was extremely conserved (Appendix A). Leucine (Leu) had the highest amino acid frequency, while cysteine (Cys) had the lowest frequency. The high leucine frequency may be related to its critical function in photosynthesis-related metabolism [29]. The measure of codon usage bias could be helpful to explore the evolutionary patterns of lineages [30].

### 3.2. The Barcoding Investigation and Phylogenetic Analyses

SSRs have been recognized as a crucial molecular marker in population genetics and conservation biology [31,32]. A total of 775 SSRs were identified in the *Vanda* plastome, with 66.32% of them being mono-nucleotide repeats, which are mainly located in the intergeneric regions of the LSC (Figure 3). The majority of the nucleotides consisted of A/T motifs rather than C/G motifs, potentially attributed to the A/T-rich plastome structure, which was consistent with the results of previous studies [33,34].

In this study, five non-CDSs regions (*clpP*-*psbB*, *rps15*-*ycf1*, *trnN^GUU^*-*rpl32*, *trnS^GCU^*-*trnG^GCC^*, and *trnT^UGU^*-*trnL^UAA^*) and six CDSs (*ccsA*, *psbI*, *rpl22*, *rpoA*, *rps15*, and *ycf1*) were selected as candidate barcodes (Figure 4). Similarly, numerous plastome comparative studies within orchids also detected some hypervariable hotspots for the species diversity groups [35,36]. Our findings can serve as specific DNA markers for species identification in *Vanda s.l*.

Plastomes provide abundant genetic information to reveal the phylogenetic relationships within a genus [37]. Previous research about the phylogenetic relationships of *Vanda* partly resolved the interspecific relationships based on several DNA markers [9,10,11,38]. In this study, our phylogenetic results based on complete plastomes (excluding one IR) and four other datasets strongly supported that *Vanda s.l.* was a monophyletic group, which is consistent with previous studies [9,10,11]. In addition, the complete plastome perfectly resolved the interspecific and infrageneric relationships of *Vanda s.l*. Among the seven sampled sections of 13 morphological sections proposed in Gardiner et al. [3], our phylogenetic results indicate that five sections were recognized, and two sections (sect. *Longicalcarata* and sect. *Cristatae*) were not monophyletic. The stem of *Vanda s.l.* diverged following the late Miocene, and rapidly diversified following the Pliocene [16]. It is acknowledged that the phylogenetic relationships among the radiated groups were difficult to resolve with several markers. For instance, compared with the results of a few DNA markers, the complete plastome, except in the case of IR, was successfully used to clarify the interspecific relationships of *Holcoglossum*, which diverged and diversified following the latest Miocene [12,13]. The plastomes also showed their power in reconstructing phylogenetic relationships of other recently diverged non-orchid lineages, such as *Eriocaulon* [14] and *Lagerstroemia* [15]. Therefore, we posit that the plastome could be valuable for exploring the phylogenetic relationships of *Vanda s.l.*, and may also be useful for other lineages that have recently undergone rapid radiation.

## 4. Materials and Methods

### 4.1. Sampling and Sequencing

In this study, five *Vanda* species, representing three sections (sect. *Ascocentrum*, sect. *Cristatae* and sect. *Longicalcarata*), were collected from Xishuangbanna Tropical Botanical Garden, Chinese Academy of Sciences. Eight published plastomes of *Vanda* were downloaded from GenBank and the annotations were updated. In total, 13 species were sampled to represent seven sections of *Vanda* as accurately as possible. Additionally, two closely related species (*Holcoglossum quasipinifolium* and *Luisia morsei*) were selected as outgroups based on Zhang et al. [16]. The detailed sampling information was listed in Appendix A.

Using the modified CTAB method [39], we extracted the total DNA of five sampled *Vanda* species from silica gel-dried leaves, respectively. Then, following the manuals of NEB Next^®^ Ultra DNA Library Prep Kit (NEB, Ipswich, MA, USA), the libraries for paired-end 150 bp sequencing were constructed using an Illumina HiSeq 2000 platform. Finally, around 4 Gb of raw data was generated for each sampled species.

### 4.2. Assembly and Annotation

After the assessment of the quality of raw sequence reads using FastQC v0.11.9 [40], we filtered the low-quality reads by Trimmomatic v0.39 [41], and assembled the clean reads using GetOrganelle v1.7.3.2 [42]. Then, to evaluate the completeness of the assembled plastomes, we checked and visualized them in Bandage v0.7.1 [43]. Finally, annotation was conducted using PGA software [44] and manually adjusted by Geneious v9.05 [45]. The plastome of *Vanda brunnea* (NC_041522) was selected as reference. The gene map was plotted by OGDRAW (https://chlorobox.mpimp-golm.mpg.de/OGDraw.html, accessed on 15 July 2024).

### 4.3. Sequence and Structure Divergence Analyses

The online program mVISTA [46] (https://genome.lbl.gov/vista/index.shtml, accessed on 15 June 2024) was used to visualize the sequence divergence of *Vanda* with the plastome of *Holcoglossum quasipinifolium* (NC_041516) as a reference. We analyzed the expansion and contraction of the IR boundary using IRscope v3.1 [47]. Possible structure divergence (rearrangements and inversions) of plastomes were detected using the Mauve v1.1.3 [48] plugin in Geneious v9.05 [45]. Moreover, the IRa region was removed to avoid the overrepresentation of the inverted repeats. To further detect hypervariable regions within *Vanda*, the Pi values of complete plastomes and 68 CDSs of 13 *Vanda* species were estimated using DnaSP v6 [49]. The window length was set to 100 sites with the step size of 25 sites.

### 4.4. Repetitive Sequence and Codon Usage Analyses

The SSRs were detected using the online program MISA [50] (https://webblast.ipk-gatersleben.de/misa/, accessed on 17 June 2024), and the parameters for SSR motifs were set to 10, 5, 4, 3, 3, and 3 nucleotide repeats for mono-nucleotide, di-nucleotide, tri-nucleotide, tetra-nucleotide, penta-nucleotide, and hexa-nucleotide, respectively. Four long repeat types in 13 plastomes of *Vanda* were determined using the online program REPuter (https://bibiserv.cebitec.uni-bielefeld.de/reputer, accessed on 17 June 2024), including F (forward), P (palindrome), R (reverse), and C (complement) repeats. The maximum and minimum repeat sizes of long repeats in the REPuter program were set to 50 bp and 20 bp, respectively, with the Hamming distance as 3. In addition, RSCU values and amino acid frequencies were calculated using MEGA v11 [51]. Protein-coding regions in the IRa region were also eliminated from the codon usage analyses to avoid overrepresentation.

### 4.5. Phylogenetic Analyses and Measurement of Divergence Variables

The phylogenetic relationships of *Vanda* were reconstructed based on the following five matrices: (1) the complete plastome sequences (excluding IRa), (2) concatenation of 68 CDSs, (3) LSC region, (4) hypervariable regions of the whole plastomes, and (5) hypervariable regions of 68 CDSs. We removed the IRa region from the complete plastome and CDSs to avoid the overrepresentation of the inverted repeats. Five matrices with 13 *Vanda* species and two outgroups were aligned using MAFFT v7 [52] and manually adjusted in BioEdit v7.0 [53]. After the selection of the best nucleotide substitution model for each matrix using jModelTest2 [54], respectively, ML analyses were carried out in RAxML v8.2.12 [55] with the best-fit model. Moreover, BI analyses were conducted in MrBayes v3.2.6 [56] for 10,000,000 generations and sampled every 1000 generations. After the assessment of the convergence in Tracer v1.7 [57], we obtained the majority rule consensus tree with the first 25% samples as “burn-in”. Finally, the phylogenetic trees were plotted using FigTree v1.4.4 (http://tree.bio.ed.ac.uk/software/Figtree/, accessed on 22 June 2024).

### 4.6. Character Reconstruction of Ndh Gene Status in Vanda

For all 13 *Vanda* species and two outgroups in our dataset, all 11 *ndh* genes were scored as lost and pseudogene. The character state reconstruction was conducted in Mesquite v3.51 (http://www.mesquiteproject.org, accessed on 12 August 2024) using the parsimony method. The best ML topology was selected as the input tree.

## 5. Conclusions

In the present study, we reported five *Vanda* plastomes (*V. ampullaceum*, *V. alpina*, *V. coerulea*, *V. cristata*, and *V. pumila*). The plastome characteristics and results of the comparative analyses indicate that the gene content and genomic structure of sampled *Vanda* plastomes were highly conserved, with the only difference was a large inversion from *trnG^GCC^* to *ycf3* in the *V. coerulea* plastome. All *ndh* genes were found to be lost or pseudogenized. According to the ranking of Pi values, five non-CDSs (*clpP*-*psbB*, *rps15-ycf1*, *trnN^GUU^*-*rpl32*, *trnS^GCU^*-*trnG^GCC^*, and *trnT^UGU^*-*trnL^UAA^*) and six CDSs (*ccsA*, *psbI*, *rpl22*, *rpoA*, *rps15*, and *ycf1*) were selected as candidate barcodes. The phylogenetic topologies based on complete plastome sequences (excluding one IR) showed the highest power in resolving the interspecific relationships of *Vanda*. Our results found that plastome can offer valuable insights into the phylogenetic relationships of recently diverged lineages.

## Figures and Tables

**Figure 1 ijms-25-09538-f001:**
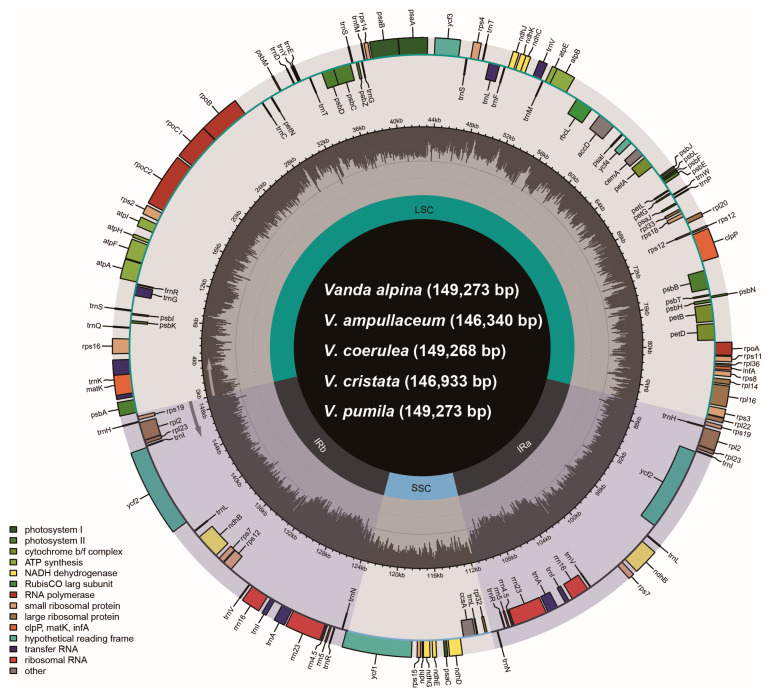
The gene map of five *Vanda* plastomes. The darker gray in the inner circle corresponds to the GC content. The IRa/b, LSC, and SSC are shown inside the GC content.

**Figure 2 ijms-25-09538-f002:**
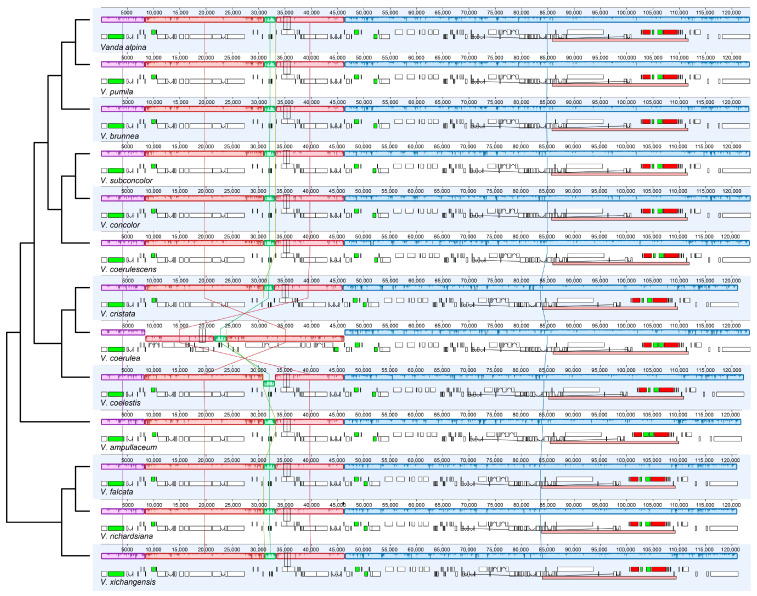
Visualization of the collinearity. Structural alignment of 13 *Vanda* plastomes using progressive Mauve. The tree topology and species order were consistent with that in Figure 5. The lines linking the collinear blocks represent homology between different genomes. Numbers on the upper *x*-axis are genome map coordinates in kilobases (kb).

**Figure 3 ijms-25-09538-f003:**
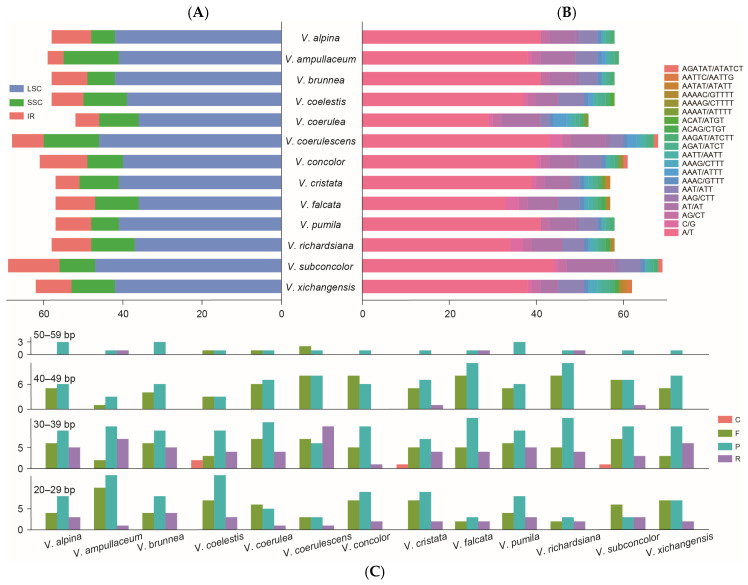
Summary of the simple sequence repeats (SSR) and long repeat sequences of 13 *Vanda* plastomes. (**A**) SSR distributions in the LSC, SSC, and IR regions. (**B**) Number of different types of SSRs identified in plastomes. (**C**) Number of long repeats by sequence length (F: forward repeats; R: reverse repeats; P: palindromic repeats; C: complementary repeats).

**Figure 4 ijms-25-09538-f004:**
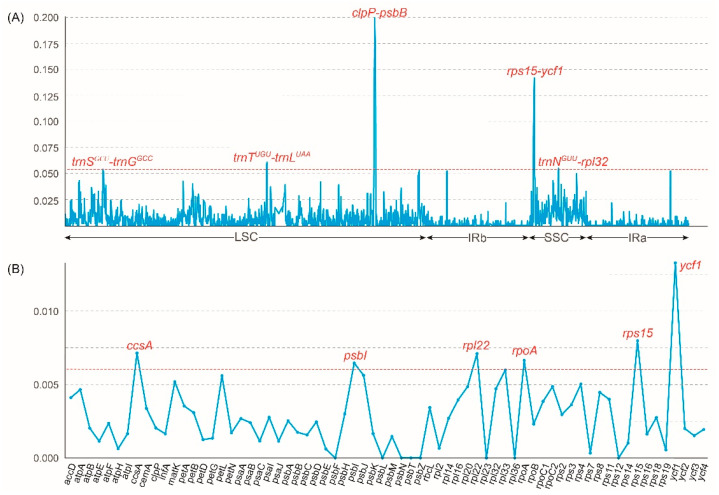
Sliding window test of nucleotide diversity (Pi) in the *Vanda* plastomes. (**A**) The nucleotide diversity of the complete plastomes showing five mutation hotspot regions (dashed line marked the Pi > 0.54). The *x*-axis represented the position of the midpoint of a window, while the *y*-axis represents the Pi value of each window. (**B**) The nucleotide diversity of 68 CDSs showing six mutation hotspot regions (dashed line marked the Pi > 0.06).

**Figure 5 ijms-25-09538-f005:**
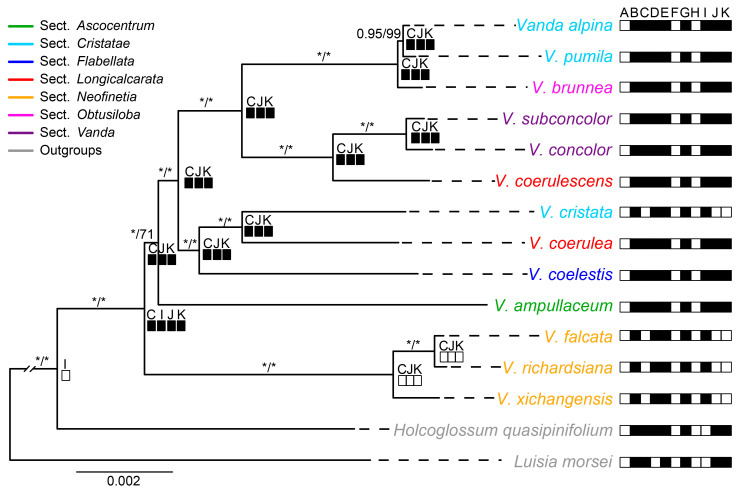
Maximum likelihood phylogenetic tree of *Vanda* based on whole plastomes with one IR and reconstruction of *ndh* gene loss and pseudogenization. The numbers above represent the supporting values of Bayesian inference and maximum likelihood analyses, respectively, and the star reflects that the supporting value is equal to 100%. The black box represents the pseudogene of the *ndh* gene, and the blank box represents the absence of the *ndh* gene.

**Table 1 ijms-25-09538-t001:** Summary of plastome characters among 13 *Vanda* species.

Taxa	Length (bp)				GC Content (%)	Gene Numbers			
	Plastome	LSC	IR	SSC		Total	CDS	tRNA	rRNA
*V. alpina*	149,273	85,819	25,860	11,734	36.7	120	74	38	8
*V. ampullaceum*	146,340	85,436	24,523	11,858	36.5	120	74	38	8
*V. brunnea*	149,216	85,783	25,860	11,713	36.7	120	74	38	8
*V. coelestis*	148,073	85,105	25,772	11,424	36.8	120	74	38	8
*V. coerulea*	149,268	85,982	25,787	11,712	36.6	120	74	38	8
*V. coerulescens*	149,410	85,954	25,965	11,526	36.7	120	74	38	8
*V. concolor*	149,474	85,678	25,897	12,002	36.6	120	74	38	8
*V. cristata*	146,993	83,927	25,772	11,522	36.7	120	74	38	8
*V. falcata*	146,497	83,808	25,457	11,775	36.6	120	74	38	8
*V. pumila*	149,273	85,819	25,860	11,734	36.7	120	74	38	8
*V. richardsiana*	146,498	83,809	25,457	11,775	36.6	120	74	38	8
*V. subconcolor*	149,490	85,691	25,912	11,975	36.6	120	74	38	8
*V. xichangensis*	146,681	83,920	25,505	11,751	36.6	120	74	38	8

## Data Availability

All the data are provided within this manuscript and Appendix A.

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
