# Peer review of "Plastome Evolution and Comparative Analyses of a Recently Radiated Genus Vanda (Aeridinae, Orchidaceae)"

_ijms, 2024, doi:10.3390/ijms25179538_

Round 1

Reviewer 1 Report

Comments and Suggestions for Authors

The study by Wan-Shun Lei et al. is devoted to the study of Plastome evolution and comparative analyses of Vanda (Aeridinae, Orchidaceae) , which is fundamentally important for the phylogenetic relationships of recently diverged lineages. The study was carried out at a good scientific level and can be recommended for publication.The amount of work in this manuscript is impressive and figures are of reasonable quality, but the authors need to clarify and improve some points.

In 2.1. The plastome characters of Vanda, I would like to wish that the authors should display the Table S1 just after the paragraph, which would be clear.

For figure1 and 2, they are not clear. Please change to clearer pictures.

For figure 5, I can not distinguish part A from part B, it should be clearer.

In section3 Discussion part, I think the authors should add one part  the Factors influencing the evolutionary rate of plant plastid genomes, that would be more impressive to the readers.

Comments on the Quality of English Language

Minor editing of English language required.

Reviewer 2 Report

Comments and Suggestions for Authors

The article is very descriptive and related to Vanda's chloroplast genomics. They have sequenced five new species. However, I have not found any specific research question the author focuses on in the article. 

The authors should provide these details very clearly. The results section will explain but the whole work is very descriptive. There are some lines about the phylogenetic which is very contradictory. Authors criticize previous studies for using few species in phylogeny, but their work is limited to just eight species. 

The author found the loss of ndh genes but this is not a new finding and previously described. 

The authors did not mention how the research of the current genus will be helpful after the availability of hundreds of genomes at the family level and 3 to 4 species at the genus level. 

The authors described the repeats but did not mention their role in the generation of indels and substitutions as described based on the correlations among mutations. 

Please see the following articles:

Correlations among oligonucleotide repeats, nucleotide substitutions, and insertion-deletion mutations in chloroplast genomes of plant family Malvaceae

Mutational dynamics of aroid chloroplast genomes II

Evolutionary origins of taro (Colocasia esculenta) in Southeast Asia (marker developed in this article)

The large single-copy (LSC) region functions as a highly effective and efficient molecular marker for accurate authentication of medicinal Dendrobium species

Comments on the Quality of English Language

The article should be revised carefully and the write up should be made further concise and to the point. 

Reviewer 3 Report

Comments and Suggestions for Authors

The paper presents five newly assembled plastome sequences for species in the Orchid genus Vanda and combines these data with whole plastome sequences from an additional eight Vanda species and two outgroup species to perform phylogenetic reconstruction analysis of this phylogenetically difficult group. 

The introduction provides mostly sufficient background information about the study system and the limits to knowledge that will be addressed. The introduction ends with clearly stated approaches and objectives. However, more context about the choice of five species should be given. Are they thought to represent different sections or a particular geographic region? Do they complement the existing plastomes in other ways? The results are well organised and mostly clearly presented. There are some errors with the presentation of phylogenetic results as the text explanation contradicts Figure 5B in several aspects. The discussion summarises the results again but does add some comparison to other studies of Orchids. The organisation is logical and relates well to the paper's stated aims. The discussion ends well by considering the application of whole plastome sequencing to other phylogenetically difficult groups. The methods are concise and mostly sufficient, but more sampling information needs to be provided. Where did the samples come from, how were they morphologically verified, and are linked botanic garden or herbarium samples available for independent verification?

The language should be revised carefully. I give several specific suggestions for clarifications below. 

Overall, this manuscripts contributes to the field of molecular phylogenetic reconstruction of Orchids. However, there are several important issues, primarily related to sampling information, but also the reasons for the choice of 68 CDSs for tree construction, and not validating the identified hypervariable regions for phylogeny reconstruction. These issues need to be addressed before the manuscript could be published. 

Specific comments

L18 Spell out CDS acronym at first use

L21 Does "compared to 68 CDSs" mean that you built a separate phylogeny for these data, or that the tree that you describe was built with these? If the latter rewrite as "using 68 CDSs". Do the recommended 10 CDSs give the same phylogenetic pattern. I think this is important to know as part of the recommendation for their use. 

L34-35 Rewrite "morphologically close to" as "morphological similarity to"

L36 Rewrite "transferring" as "transferred"

L46 Remove "been" from "has been successfully"

L48 Rewrite "diverged radiated" as "divergently radiated"

L66 Specify what you are comparing to when you state the structure is typical.

L72 Is 120 genes the sum across the 13 plastomes or the number of genes per plastome?

L95-96 The lines in Figure 2 seem to represent rearrangements in the figure. Are not the plastomes mostly homologous across their whole length?

L118 What determined the choice of these 68 CDSs out of all identified CDSs?

L150-151 Explain the choice of outgroups that were used to confirm monophyly. 

L150 Explain the exclusion of one IR.

L151 Explain pp-BI and BS-ML acronyms at first use. 

L155 I struggle to distinguish the colours of the legend to Figure 5 but sect. Obtusiloba seems to be only represented by a single species (V. brunnea) meaning that its monophyly cannot be assessed. 

L156-158 The branch support values shown on Figure 5B are all high in contrast to the text statement. 

L174-175 I recommend to elaborate this statement by linking to the function of ndh genes.

L207 Figure 4 shows five not four hypervariable non-CDS regions. 

L209-210 Similar to my earlier point, i think you should perform phylogenetic analysis for these 10 (11) concatenated regions to confirm their value in phylogenetic reconstruction of Vanda.  

L224-225 Specify that you are referring to the difficulties in determining monophyly of different sections here. Give an approximate timeline in Ma to give more context to the reader.

L237 Where were the leaf samples obtained from? Ideally there should be linked herbarium samples to allow independent verification of morphological species assignment. I do not have access to supplementary information to view Table S2 sample data for confirmation. 

L255-256 Explain why one IR copy needed to be removed for collinearity analysis. 

L262-265 I do not follow the choice of 10,5,4,3,3,3 nucleotide repeats for the SSR search. Why skip 9-6 repeats, why repeat 3 repeat analysis three times?

L274-275 Similar to a previous comment, explain why one IR was excluded. Explain the choice of 68 CDSs from all the CDS identified.

L284 I suggest to mention how the trees were plotted.

Comments on the Quality of English Language

Please see specific comments for some editing suggestions. 

Round 2

Reviewer 2 Report

Comments and Suggestions for Authors

The authors did a good job in the revision. However, still, the article does not answer the main question: How the current article is scientifically important? There are still many issues in the articles:

What is the main research question of the article? The manuscript still needs significant improvement in the write-up: 

For example: 

line 78; 

For example, lines 121-126 are not even acceptable at the language level, not at the scientific level. The authors want to mention the preference for codons ending with A/T at 3' end. However, they failed to do so.

 The total number of genes was 74; However, they concatenated 68 protein-coding genes and determined the codon usage pattern. 

Results should be in the past tense, but they mentioned results in the present tense. for example, in line 69; 

line 86: please check: The IRa and IRb, LSC and SSC are indicated the outside of the GC content; 

Improve the description of Figure 2.

Please look at the whole article for such issues.  

Line 328-333: The authors mentioned the methodology for correlation analysis. The authors should either use the appropriate method described in the articles mentioned by me.  Authors can mention the role of repeats (the repeats author identifies based on REPuters, not based on MISA). So,  I will suggest removing this part, including the method section and results. I noticed that you found the correlations to be just 0.229, etc. This shows weak correlations, not strong correlations. So, I will suggest that you do some qualitative analysis or avoid them and provide a comment based on previous articles in the discussion. I provided the comment because you did the analysis but did not discuss this point in a descriptive article, which is a little vague. 

You can provide a comment like below: 

Repeat analysis and utilization of oligonucleotide repeats as proxy to identify polymorphic loci

Please see for detail the article: 

Comparative plastome analysis of Blumea, with implications for genome evolution and phylogeny of Asteroideae

You need to look at the article in more detail and make corrections. 

One point I want to mention is the similarity index of the article. The editor can consider this point more. 

Comments on the Quality of English Language

The English write-up is overall not bad but some common mistakes and inappropriate structure of sentences exist in the text that need approval. 

Reviewer 3 Report

Comments and Suggestions for Authors

The authors have made extensive revisions to the paper including some additional recommended analyses that have helped refine their study conclusions. All of my major suggestions have been addressed adequately in the responses. I am now satisfied to endorse this submission.

With reference to my specific recommendations, the different subsets of sequence data used for phylogenetic analysis is now clearly explained in the abstract and other parts of the paper. The sampling information has been improved and is now satisfactory. The extra suggested phylogenetic analyses have been performed leading to some reinterpretation of the results that whole plastomes provide the best species resolution for Vanda. This improves the detail and interpretation of the study. The revised phylogeny figure is simplified and easier to interpret. Extra background information on ndh gene function adds interesting context. Extra discussion of related findings in the field also helps strengthen the interpretation of the results. Many other changes have been made throughout the manuscript to improve the language.

Specific comments

L121-122: I think the reason for choosing 68 CDSs should be added here to the text.

L275 Typo: replace "closed" with "closely related"

Round 3

Reviewer 2 Report

Comments and Suggestions for Authors

The article can be accepted. The author needs to look at the article again in the proofreading stage. I just noticed some minor errors. For example, However, was "owever" in one place. 

I appreciate the authors for their hard work. 

Regards

Comments on the Quality of English Language

The language is OK.